# Menopausal Status Contributes to Overall Survival in Endometrial Cancer Patients

**DOI:** 10.3390/cancers15020451

**Published:** 2023-01-10

**Authors:** Bartłomiej Barczyński, Karolina Frąszczak, Artur Wnorowski, Jan Kotarski

**Affiliations:** 11st Chair and Department of Oncological Gynaecology and Gynaecology, Medical University in Lublin, 20-081 Lublin, Poland; 2Department of Biopharmacy, Medical University in Lublin, 20-081 Lublin, Poland; 3Independent Laboratory of Cancer Diagnostics and Immunology, Medical University in Lublin, 20-081 Lublin, Poland

**Keywords:** endometrial cancer, overall survival, prognosis, Cox proportional hazards model, menopause

## Abstract

**Simple Summary:**

Endometrial cancer is a leading cause of female genital tract malignancies, and its incidence has been rising in recent years. Despite relatively good prognosis, a noticeable number of patients eventually die within five years of diagnosis. Thus, one of the most common questions addressed to oncologists by patients and their families refers to prognosis. Our study was performed to assess the influence of some conventional clinicopathological features on overall (all-cause) survival of endometrial cancer patients. We have proved that some characteristics related to menopause status (i.e., time from last menstruation and postoperative FSH concentration) may strongly correlate with patients’ survival. Further research on various risk factors will provide us with improved and more personalized care for cancer patients.

**Abstract:**

Endometrial cancer is the most common female genital tract malignancy in developed countries that occurs predominantly in postmenopausal women. The primary objective of our research was to investigate whether menopause status together with selected conventional prognostic indicators may contribute to overall (all-cause) survival in endometrial cancer patients. For this purpose, we applied the Cox proportional hazards regression model. Patients in advanced FIGO stage showed a relatively poor survival rate. The time since last menstruation and postoperative FSH concentration were identified as unfavorable prognostic factors in our model. Additionally, age at diagnosis, BMI value, adjuvant treatment (brachytherapy), and parity showed no impact on survival. To our knowledge, this is the first study to report a prognostic model for endometrial cancer including exact time from last menstruation as one of the prognostic variables. Due to the fact that there are no stratifying systems to reliably predict survival in patients with endometrial cancer, there is a strong need to revise and update existing models using complementary prognostic indicators. Collection of precise data on various risk factors may contribute to increased accuracy of artificial intelligence algorithms in order to personalize cancer care in the near future.

## 1. Introduction

Endometrial cancer is the most common female genital tract malignancy in developed countries, and its incidence rate has been increasing during recent decades, particularly among postmenopausal women. Despite the generally good overall prognosis for uterine cancer, approximately 20% of patients are expected to eventually die from the disease [1]. Several factors have been shown to influence the risk of developing endometrial cancer, including age, obesity, diabetes mellitus, reproductive features (e.g., nulliparity), diet and exercise, as well as the use of certain medications (unopposed estrogen hormonal therapy, tamoxifen). However, the most important prognostic factors regarding cancer-specific and overall survival include older age at diagnosis (>65), together with surgical FIGO stage, myometrial invasion, histological type, differentiation grade, lymphovascular space invasion, and nodal status [1,2,3]. In 2013, The Cancer Genome Atlas Research Network classified endometrial cancers into four categories: POLE ultramutated, microsatellite instability hypermutated, copy-number low, and copy-number high [4]. Based on this classification, a new Proactive Molecular Risk Classifier for Endometrial Cancer (ProMisE) classifying system was developed. This system provides independent prognostic information beyond established clinicopathologic risk factors available at diagnosis [5]. Tumors with POLE exonuclease domain mutations had the most favorable prognosis, while those with p53 null/missense mutations had the worst [5]. However, due to some logistic and financing issues, the use of multiple molecular classifiers remains challenging and the frequency of their application in all published studies on endometrial cancer cases is low, ranging from 3% to 5% [6]. Collaborative studies from across the world are working to define pragmatic assays, improve risk stratification systems by combining novel molecular features and traditional clinicopathological parameters, and determine how molecular classification can be optimally utilized to direct patient care and predict survival [6]. Thorough identification of every individual prognostic indicator is especially important in the era of implementation of artificial intelligence techniques in clinical practice, improving risk stratification and surveillance management of cancer patients [7].

Endometrial cancer is a disease that occurs predominantly in postmenopausal women, with 75–80% of women being postmenopausal at the time of diagnosis [8]. Menopause has been shown to occur usually between the ages of 40 and 58, with an average age of 51 years [9]. Other authors more precisely define that the majority of women enter natural menopause between the ages of 49 and 52 years [10]. Sometimes, defining menopause might be quite challenging in a clinical setting. To date, there is no sensitive screening tool for predicting the age of natural menopause. Thus, most clinical registries define the menopause status of female patients in regard to their last menstrual period and/or according to specific changes in hormone levels, including anti-Müllerian hormone, inhibin A or inhibin B, estradiol, follicle-stimulating hormone (FSH), and luteinizing hormone (LH). Although some authors question routine use of hormonal tests to diagnose menopause [11], many clinical protocols implement FSH screening serum as a confirmation of patients’ menopause status [12,13,14].

In the present study, we decided to assess the influence of some conventional clinicopathological features on overall (all-cause) survival of endometrial cancer patients. We examined the association of age, FIGO stage, differentiation grade, body mass index, parity, first and last menstruation, selected serum markers (CA125 and HE4), selected serum hormones (LH and FSH), and type of adjuvant treatment on overall survival. The main aim of our research was to investigate whether menopause status together with selected conventional clinicopathological features may play an important prognostic role in patients with endometrial cancer.

## 2. Materials and Methods

### 2.1. Patients’ Characteristics

The study cohort included individuals operated on due to endometrial cancer in the 1st Department of Oncological Gynaecology and Gynaecology of the Medical University in Lublin from September 2013 to April 2016. Only cases with endometrioid type of endometrial adenocarcinoma, postoperatively staged as FIGO I–IV, were included in the study. All postoperative pathology reports, including type of cancer and differentiation grade, were confirmed by two independent pathologists. All patients had undergone standard surgical treatment, including total hysterectomy and bilateral salpingo-oophorectomy ± pelvic lymphadenectomy via laparotomy or laparoscopy, and standard adjuvant treatment (brachytherapy, brachytherapy + external beam radiotherapy or radiotherapy + chemotherapy), if needed according to European Society of Gynaecologic Oncology guidelines.

Complete preoperative clinical data regarding age, parity, body mass index, and first and last menstruation were collected either directly from patients or from medical hospital records. Data on preoperative serum concentrations of follicle-stimulating hormone (FSH), luteinizing hormone (LH), cancer antigen 125 (CA125), and human epididymis protein 4 (HE4) were obtained. FSH and LH concentrations were also assessed 4 weeks after surgery. All patients were contacted every six months to assess disease-free survival, recurrence, and overall (all-cause) survival. Last contact was performed in June 2022. Our retrospective single-institution study design was approved by the Medical University in Lublin Ethics Committee (KE-0254/273/2020). All patients signed an informed consent form.

### 2.2. Laboratory Tests

Peripheral blood was acquired at hospital within 1 day before surgery (CA125, HE4, FSH, and LH) or 4 weeks after surgery during a routine postoperative control visit. All blood serum tests were performed in the hospital laboratory, using routine hospital procedures and standard assays, i.e., electrochemiluminescence immunoassays “ECLIA” Elecsys CA 125 II (Roche Diagnostics, Basel, Switzerland) for CA125, Elecsys HE4 (Roche Diagnostics) for HE4, chemiluminescent microparticle immunoassays “CMIA” Architect LH 2P40 (Abbott Laboratories, Chicago, IL, USA) for LH, and Architect FSH 7K75 (Abbott Laboratories) for FSH, respectively.

### 2.3. Statistical Analysis

The Cox proportional hazards model was used to examine potential predictor variables for all-cause mortality in endometrial cancer patients. The following predictor variables were selected for the model: age at diagnosis (years), FIGO stage (IA, IB, and II–IV), BMI value (kg/m^2^), adjuvant therapy (no adjuvant treatment or brachytherapy), parity (number of offspring), time since last menstruation (years), and postoperative FSH concentration (mIU/mL). Age, BMI, parity, time since last menstruation, and FSH concentration were continuous variables. Remaining variables were categorical.

Akaike’s Information Criterion (AIC) was used to assess whether the specified model with selected predictor variables estimated hazard rate better than the null model. Our model scored an AIC value of 175.7, which is less than the value of 184.4 calculated for the empty model. Scaled Schoenfeld residuals were plotted against time and no clear trend was observed, indicating that the proportional hazards assumption of the analysis was not violated (Appendix A). Deviance residuals vs. hazard ratio graph were used to examine for any potential outliers (Appendix A). No extremely large or extremely small values representing potential outliers were observed. Deviance residuals were plotted against each of the covariates in the model, and no clear trends were observed (Appendix A). This indicates that the linearity assumption of the selected covariates has not been violated. The conducted quality check of the analysis indicates that the constructed model is valid.

Curve comparison for Kaplan–Meier survival analysis was conducted using log-rank (Mantel–Cox test).

All analyses were performed using GraphPad Prism 9.4.1 (GraphPad Software). The significance level (*α*) was set to 0.05.

## 3. Results

### 3.1. Characteristics of the Study Population

Our final study cohort included 92 endometrial cancer patients. The clinical characteristics of the patients, with 21 (22.8%) deaths, are summarized in Table 1. The results of the CA125, HE4, FSH, and LH measurements in the patients’ serum are summarized in Table 2.

### 3.2. Effects of Prognostic Factors on Overall Survival

We applied the Cox proportional hazards model to analyze the all-cause mortality with the proportional hazards assumption. Hazard ratios and respective 95% confidence intervals, along with corresponding *p*-values, are listed in Table 3. The expected hazard for patients characterized by FIGO II+ (II-IV) stage was 10.77 times higher in comparison to FIGO IA patients, holding other variables constant (Figure 1A). Time since last menstruation (HR = 1.151) and postoperative FSH concentration (HR = 1.020) were also identified as unfavorable prognostic factors in our model. In more detail, the time period of five years after menstruation increased the hazard rate by 1.151^5^ = 2.020. Correspondingly, increases of 10 and 15 years led to the increase the hazard rate by 4.081 and 8.244, respectively (Figure 1B). For 30 and 70 mIU/mL of FSH in a patient’s serum, the hazard rate increased to 1.811 and 4.000, respectively (Figure 1C). Additionally, age at diagnosis, BMI value, adjuvant treatment, and parity showed no statistically significant impact on survival.

To highlight the predictive effect of the time since last menstruation, we compared the estimated survival for patients at the age of 60, 63.5 (median), and 67. For each age group, three curves depicting the survival of individuals with either 0, 11 (median) or 22 years of postmenopausal life were plotted (Figure 2). Survival curves for patients of different age but same time since last menstruation clustered together, demonstrating the detected predictive prognostic significance of time since last menstruation.

Finally, we divided the patients into two age groups: (a) younger patients that were 45–64 years old (Figure 3A), and (b) older patients that were 65 and older (Figure 3B). Then, we performed the Kaplan–Meier survival analysis, taking into account the time since last menstruation, independently for each age group. In both age groups, the survival curves for patients with shorter postmenopausal life at diagnosis displayed significantly better survival (Figure 3), further demonstrating the prognostic value of time since last menstruation.

## 4. Discussion

One of the most common questions addressed to oncologists by patients and their families refers to cancer prognosis. Although the general prognosis for endometrial cancer is better than that of other gynecological malignancies, many patients still recur within 5 years and die despite the application of adjuvant therapy. Early endometrial cancer (FIGO stage I and II) recurrence rates range from 5% to 15%, and three quarters of the relapse patients are reported during the first 3 years after primary treatment [15,16]. Undoubtedly, the sooner neoplasm relapse occurs, the shorter the overall survival rate. In a recently published study by Dou et al., the overall 5-year survival of early-relapse patients (within the first 36 months after primary treatment) with FIGO stage I/II endometrial cancer (EC) was 60.3%, while the overall survival of late-relapsed individuals (>36 months after primary treatment) was closer to 81% [16]. According to current recommendations, patients with low-risk endometrial cancer do not require adjuvant treatment due to the low risk of recurrence. However, patients with intermediate risk should receive adjuvant radiation therapy to reduce local recurrence, although this management did not improve overall survival in cohorts of the PORTEC study [17]. Thus, a lot of effort is now being put into searching for endometrial cancer prognosis-related clinical characteristics and demonstrating endometrial cancer prognostic models with high accuracy to introduce valuable counseling tools for evaluating reliable individualized prognosis, thereby improving the overall survival of endometrial cancer subjects. Previous studies have proved that the clinical variables related to endometrial cancer prognosis include predominantly FIGO stage, differentiation grade, histological type, age at diagnosis, depth of muscular invasion, lymphovascular space invasion, and lymph node metastasis [1,2,3,18]. Novel prognostic models based on clinical characteristics analyze such factors as BMI, visceral adiposity index, type 2 diabetes mellitus, and menopause status [19,20,21].

In the present study, we applied Cox proportional hazards regression analysis to relate several risk factors, considered simultaneously, to survival time. In this model, the measure of effect is the hazard rate, which is the risk of failure, given that the patient has survived up to a specific time. Our model included eight different variables: age at diagnosis, FIGO stage (IA, IB and II–IV), BMI value, adjuvant treatment (brachytherapy), parity, time since last menstruation, and postoperative FSH concentration. In our Cox analysis, FIGO II+ stage showed the strongest correlation to shorter overall survival (*p* < 0.0001), which confirmed that FIGO staging not only acts as a guideline for treatment but also has robust potential as an indicator for prognosis. We also found that in our study cohort, neither age at diagnosis nor BMI value, adjuvant brachytherapy, or parity were negatively correlated to survival (*p* > 0.05). What is most interesting is that variables related to menopause status, i.e., time from last menstruation, and postoperative FSH concentration were negatively correlated to endometrial cancer survival (*p* < 0.05).

To our knowledge, this is the first study to report a prognostic model for endometrial cancer including exact time from last menstruation as one of the variables. Data on the prognostic value of menopause status are still not consistent enough to provide reliable conclusions. Most analyses evaluating the prognostic significance of menopause status in endometrial cancer define patients as pre- or postmenopausal, regardless of time of last menstruation [22,23,24]. In a study from Li et al., postmenopausal status served as an independent factor for poor prognosis, whereas according to Huang et al., menopause showed no such influence, and both studies were performed in large cohorts of Chinese women [23,24]. Similarly, Wang et al. did not identify menopause status as an independent risk factor of worse overall survival in endometrial cancer patients [25]. In a study from Nakanishi et al. performed on Japanese patients, women who were operated on due to endometrial cancer at least five years after last menstruation had a far worse prognosis than younger individuals [26]. Completely opposite results were published by Gottwald et al., who concluded that women who had undergone menopause at a relatively younger age had much shorter overall survival [27].

Another significant clinical problem is related to undertreating occult endometrial cancer in patients with endometrial hyperplasia. Endometrial hyperplasia does not require the same aggressive treatment approach as endometrial cancer and is usually managed by either progestin hormonal therapy or a simple hysterectomy [28]. Many such patients with occult endometrial cancer are subjected to incomplete surgeries and end up being classified as incompletely staged cancer, ultimately leading to a worse prognosis [28]. According to the literature, one of the most important clinical predictors of concurrent endometrial cancer is menopause status. In a multivariate model presented by Rajadurai et al., postmenopausal status of patients with atypical hyperplasia was one of the significant predictors of endometrial adenocarcinoma in the Western Australia population [29]. Postmenopausal status and nulliparity were also independent significant factors for concurrent endometrial cancer in atypical hyperplasia patients in a study of Karakas et al. [30]. Therefore, exact menopause status seems to be a key clinical factor in predicting endometrial cancer prognosis and survival in proportional hazards models studies.

Endometrioid adenocarcinoma is traditionally classified as Bokhman’s type 1, estrogen-dependent endometrial cancer. It was proved that the concentrations of female sex hormones, such as estradiol, FSH, and LH, differ in pre- and postmenopausal women with endometrial cancer [31,32]. It is also commonly known that surgical menopause causes a significant rapid drop in systemic estradiol level and an increase in LH and FSH levels. In our research, we observed similar changes in LH and FSH levels as a physiologic consequence of surgical procedures performed in premenopausal endometrial cancer patients, and no such changes in LH and FSH concentrations in postmenopausal individuals. However, there were no significant changes in overall preoperative and postoperative LH and FSH concentrations in regard to menstrual status. In our initial proportional hazards model, we included all four hormonal variables, i.e., preoperative and postoperative LH and FSH concentrations, respectively. However, postoperative FSH levels served as the only hormonal variable to have significant prognostic value in our model.

Some limitations of our study need to be discussed. First of all, our research was a retrospective single-institution cohort study, lacking the validation of prospective multi-center studies. Furthermore, survival studies in the post-TCGA era should take into consideration new risk classifying systems (e.g., ProMisE). However, application of TCGA risk stratification methodology is still too expensive and burdensome for widespread implementation into routine clinical everyday practice. Additionally, as the TCGA classification was the result of unsupervised clustering of genetic divergences of small and clinically heterogeneous cohorts, it is still insufficiently powered to address clinical utility [6]. More likely, the prognostic ability of novel molecular classifiers should be further improved by integrating them with previously existing prognostic clinicopathological variables. Furthermore, it has to be pointed out that one of the most important aspects of our study is the fact that it covers practical aspects/elements which might become predictor factors (i.e., menopausal status and FSH assays) that are easy-to-use tools in everyday clinical practice.

## 5. Conclusions

We have demonstrated here that some of the basic clinical variables related to menstrual status, such as time from last menstruation or postoperative FSH concentration, may contribute to overall survival in endometrial cancer patients. Up to now, there have been no risk-stratifying systems showing high accuracy to reliably predict survival of early-stage endometrial cancer patients. Therefore, there is a strong need to revise and update existing models using available complementary prognostic indicators to individualize cancer management. Collection of precise data on various clinicopathological risk factors may contribute to increased accuracy of artificial intelligence algorithms needed to personalize cancer care in the near future.

## Figures and Tables

**Figure 1 cancers-15-00451-f001:**
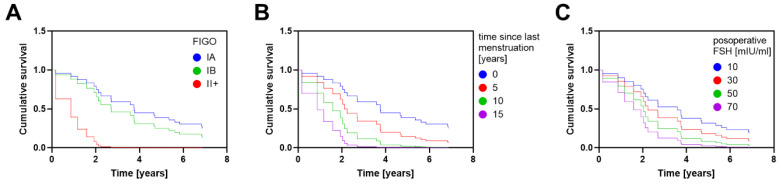
Estimated survival curves generated from the proposed Cox proportional hazards model for patients with different FIGO stage (**A**), time since menstruation (**B**), and postoperative FSH serum concentrations (**C**). For each plot, the variables not depicted on the plot were set either to zero (for continuous variables) or to the reference value (for categorical variables).

**Figure 2 cancers-15-00451-f002:**
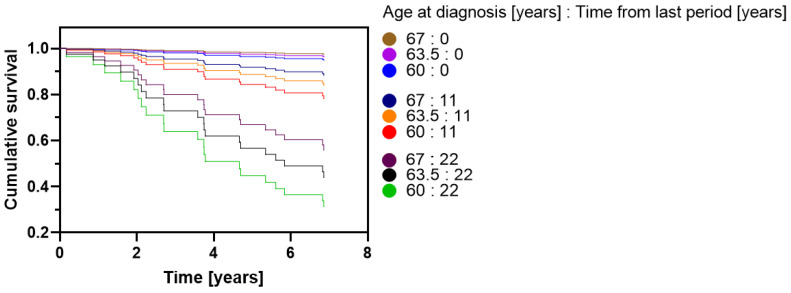
Estimated survival of patients characterized by different age at diagnosis (i.e., 60, 63.5, or 67 years) and time from the last period (i.e., 0, 11, or 22 years). The curves were generated based on the predicted baseline survival and coefficients from the Cox proportional hazards regression model. BMI and postoperative FSH level were set to median value (32 kg/m^2^ and 50.1 mIU/mL, respectively). FIGO was set to IB, and parity was set to 0. The calculations assumed no adjuvant treatment.

**Figure 3 cancers-15-00451-f003:**
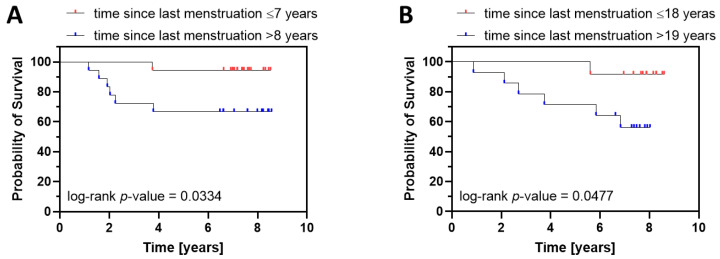
Kaplan–Meier survival analysis for younger ((**A**), 45–64 years old) and older ((**B**), 65 and more) endometrial cancer patients characterized by shorter and longer time since last menstruation. (**A**). The survival curve for patients with shorter time since last menstruation (≤7) was significantly different from the curve for patients with longer time since last menstruation (>8). Statistics: *p*-value (log-rank test) = 0.0334; HR (log-rank) 0.1404, 95% C.I. 0.03174–0.6213. (B) The same relationship was observed in older patients. Statistics: *p*-value (log-rank test) = 0.0477; HR (log-rank) = 0.1564, 95% C.I. 0.03551–0.6886. Since multivariate Cox analysis demonstrated the significant prognostic value of postoperative FSH levels and FIGO stage, patients displaying extreme FSH values (bottom 5% and top 5%) were excluded, together with FIGO IB patients.

**Table 1 cancers-15-00451-t001:** Clinicopathological characteristics of 92 endometrial cancer patients included in the study. BMI: body mass index; FIGO: the International Federation of Gynecology and Obstetrics; TAH: total abdominal hysterectomy; TLH: total laparoscopic hysterectomy; BSO: bilateral salpingo-oophorectomy; LND: lymphadenectomy.

Parameter	Value
Median (range) Age, years	63.5 (37–86)
Median age of menarche (range) Age, years	14 (10–19)
Median age of menopause (range) Age, years	52 (37–61)
Median time from menopause to cancer diagnosis (range) years	11 (0–38)
Median (range) BMI kg/m^2^	32.0 (19.6–46.6)
BMI category, n (%)	
Normal	10 (10.9%)
Overweight	25 (27.2%)
Obese	57 (62.0%)
Tumor differentiation grade, n (%)	
G1	19 (20.7%)
G2	66 (71.7%)
G3	7 (7.6%)
FIGO stage, n (%)	
IA	52 (56.5%)
IB	22 (23.9%)
II–IV	18 (19.6%)
Endometrioid histology, n (%)	92 (100%)
Surgical treatment, n (%)	
TAH/TLH + BSO	40 (43.5%)
TAH/TLH + BSO + LND	52 (56.5%)
Type of adjuvant treatment, n (%)	
None	34 (37.0%)
Brachy ± external beam radiotherapy	58 (63.0%)
No of patients who recurred, n (%)	15 (16.3%)
Overall survival rate, n (%)	
FIGO IA	46 (88.5%)
FIGO IB	18 (81.8%)
FIGO II–IV	7 (38.9%)

**Table 2 cancers-15-00451-t002:** Laboratory results of 92 endometrial cancer patients included in the study. CA125: cancer antigen 125; HE4: human epididymis protein 4; LH: luteinizing hormone; FSH: follicle-stimulating hormone.

Parameter	Value
Mean CA125 serum concentration (range) [U/mL]	232.1 (4.7–11336.0)
Mean HE4 serum concentration (range) [pmol/l]	190.3 (34.3–3137.0)
Mean preoperative LH serum concentration (range) [IU/mL]	20.3 (2.6–48.7)
Mean preoperative FSH serum concentration (range) [mIU/mL]	46.9 (1.1–97.5)
Mean postoperative LH serum concentration (range) [IU/mL]	22.4 (0.1–53.7)
Mean postoperative FSH serum concentration (range) [mIU/mL]	50.1 (1.0–116.0)

**Table 3 cancers-15-00451-t003:** Hazard ratios and *p*-values for the Cox proportional hazard regression model. An asterisk symbol (*) was used to denote the variables identified as statistically significant (*p* < *α*). BMI: body mass index; FIGO: the International Federation of Gynecology and Obstetrics; FSH: follicle-stimulating hormone.

Variable	Hazard Ratio	95% CI (Profile Likelihood)	*p*-Value
Age [years]	0.9060	0.7943 to 1.039	0.1460
FIGO [IB vs IA]	1.468	0.3410 to 5.856	0.5866
FIGO [II–IV vs IA]	10.77	3.514 to 37.95	<0.0001 (*)
BMI [kg/m^2^]	1.041	0.9559 to 1.137	0.3582
adjuvant treatment [brachytherapy vs. no adjuvant treatment]	0.9537	0.3633 to 2.581	0.9235
parity [number of offspring]	0.8941	0.6172 to 1.254	0.5326
time since last menstruation [years]	1.151	1.003 to 1.329	0.0478 (*)
postoperative FSH serum concentration [mIU/mL]	1.020	0.9997 to 1.040	0.0493 (*)

## Data Availability

The data presented in this study are available on request from the corresponding author.

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
