# Peer review of "Menopausal Status Contributes to Overall Survival in Endometrial Cancer Patients"

_cancers, 2023, doi:10.3390/cancers15020451_

Round 1
Reviewer 1 Report
Dear Author,
This is an interesting paper.
Here are my observations/questions/comments:
1. Summary – You should clearly specify your contribution to the field – “some of such characteristics”
2. Abstract – The aim should be placed before the studied parameters (lines 20-21)
3. Introduction – Line 51 – Do you mean “diabetes mellitus”?
4. Method - Ethical aspects should be placed after study design, inclusion/exclusion criteria, evaluation of the patients
5. Method – The number of patients was obtained after applying inclusion/.exclusion criteria (it is actually a part of the Results)
6. Results – Table 1 – The years in (since) menopause (or months) represents an alternative to the age of menopause for the menopausal women and should be specify in Table 1, too.
7. Title – Conclusion – Rather the time since menopause than menopausal age seems to matter (in addition to FSH). I suggest to use “menopausal status” instead of “menopausal age”
8. Line 280 – No need for “in conclusion” (it is already mentioned – line 279)
9. Discussion – One important aspect of your study is the fact that it covers practical aspects/elements that might become predictor factors like assessing the menopausal status and FSH assays which are easy-to-use tools for daily practice
Thank you very much,
Best regards,
Author Response
Dear reviewer 1, thank You for Your comments improving quality of our paper. Your suggestions were taken into consideration and changes were provided:
- Summary – You should clearly specify your contribution to the field – “some of such characteristics”
Information regarding contribution to the field has been added to Summary paragraph (line 17)
- Abstract – The aim should be placed before the studied parameters (lines 20-21)
We changed the order of the sentences in abstract paragraph according to Your suggestion (lines 21-27)
- Introduction – Line 51 – Do you mean “diabetes mellitus”?
We updated the terminology: diabetes mellitus (line 54)
- Method - Ethical aspects should be placed after study design, inclusion/exclusion criteria, evaluation of the patients
We changed the ethical issues positioning in Materials and Methods paragraph (line 117-119)
- Method – The number of patients was obtained after applying inclusion/.exclusion criteria (it is actually a part of the Results)
We slightly changed text regarding number of patients in both: Materials and Methods, and Results paragraphs (line 99-101, and 153-154)
- Results – Table 1 – The years in (since) menopause (or months) represents an alternative to the age of menopause for the menopausal women and should be specify in Table 1, too.
Data on "Median time from menopause to cancer diagnosis (range) years" were added to Table 1
- Title – Conclusion – Rather the time since menopause than menopausal age seems to matter (in addition to FSH). I suggest to use “menopausal status” instead of “menopausal age”
The title was slightly changed according to Your suggestions.
- Line 280 – No need for “in conclusion” (it is already mentioned – line 279)
The error was corrected (line 289)
- Discussion – One important aspect of your study is the fact that it covers practical aspects/elements that might become predictor factors like assessing the menopausal status and FSH assays which are easy-to-use tools for daily practice
The sentence regarding one of the most important aspects of our study was added at the end of Discussion paragraph (line 284-287).
We are grateful for careful review of entire manuscript.
Regards,
Authors of the manuscript.
Reviewer 2 Report
This article contains the relationship between many variables including time after menopause and serum FSH level and patients’ prognosis in endometrial cancer. Authors present interesting findings, however, there are a couple of questions that should be answered.
1. Authors stated that time since last menstruation is related with patients’ prognosis. However, it seems that prognostic difference between groups with time since last menstruation merely reflect patients’ age. In order to prove the prognostic significance of time since last menstruation, authors should compare the patients with different groups of time since last menstruation in a same age range.
2. It is interesting that serum FSH level has prognostic significance in endometrial cancer. It is also intriguing to see the relationship of FSH level and the histological grade of endometrial cancer.
Author Response
Dear reviewer, we would like to thank You for Your careful revision of our manuscript. We discuss Your valuable comments below:
1. Authors stated that time since last menstruation is related with patients’ prognosis. However, it seems that prognostic difference between groups with time since last menstruation merely reflect patients’ age. In order to prove the prognostic significance of time since last menstruation, authors should compare the patients with different groups of time since last menstruation in a same age range.
The authors thank the reviewer for this very stimulating question. There are a few issues that we want to address here. First, if the ‘time since last menstruation’ simply reflect patients’ ‘age’, there should be a strong and significant correlation between the two factors. And, as one could expect, ‘time since last menstruation’ and ‘age’ were well correlated (Preason r = 0.896, p‑value < 0.0001). And this makes perfect biological sense – older woman experience longer postmenopausal life.
Yet, the correlation was not perfect (r ≠ 1). This creates the possibility that the two studied factors have slightly different predictive power. And that is exactly what we saw in our Cox analysis, were ‘time since last menstruation’ (p-value = 0.0478), but not ‘age’ (p-value = 0.1460), demonstrated prognostic significance. We run Kaplan-Meier survival analysis for ‘age’ subgroups (5, and 10 years span) with different ‘time since last menstruation’ values but it resulted in extremely small n, drastically decreasing the power of the test, and making drawing any conclusions impossible. And that is exactly why we used more powerful Cox analysis in the first place.
2. It is interesting that serum FSH level has prognostic significance in endometrial cancer. It is also intriguing to see the relationship of FSH level and the histological grade of endometrial cancer.
As directed by the interest of the reviewer, we investigated the relationship between pre- and post-operative FSH levels vs. the histological grade in our group of endometrial cancer patients. There was no significant correlation between the two factors, as depicted below (please see the attachment).

Round 2
Reviewer 2 Report
This study still needs to show the prognostic significance of time since last menstruation by comparing age groups.
If the comparison of the same age range with the same group of time since last menstruation is difficult, authors need to expand the study size.
Author Response
Dear reviewer,
We would like to thank you again for your valuable comments which helped to improve this manuscript. New/corrected parts of the paper are in the track changes mode to facilitate the assessment of changes. We tried to do our best to fulfil your expectations and we hope that you will be satisfied with our corrections.
1. This study still needs to show the prognostic significance of time since last menstruation by comparing age groups. If the comparison of the same age range with the same group of time since last menstruation is difficult, authors need to expand the study size.
Response:
To highlight the predictive effect of the time since last menstruation, we compared the estimated survival for patients at the age of 60, 63.5 (median), and 67. For each age group, three curves depicting the survival of individuals with either 0, 11 (median) or 22 years of postmenopausal life were plotted (data shown on added Figure 2). Survival curves for patients of different age but same time since the last menstruation clustered together, demonstrating the detected predictive prognostic significance of time since last menstruation.
Finally, we divided the patients into two age groups: (a) younger patients that were 45-64 years old (data shown on added Figure 3A), and (b) older patients that were 65 and older (data shown on added Figure 3B). Then, we performed the Kaplan-Meier survival analysis taking into account the time since last menstruation, independently for each age group. In both age groups, the survival curves for patients with shorter postmenopausal life at diagnosis displayed significantly better survival (data shown on added Figure 3), further demonstrating the prognostic value of time since last menstruation.
Best regards, authors of the manuscript
Round 3
Reviewer 2 Report
This version of the article is well revised. They clearly show the difference of prognosis between the same age groups. It would be intriguing to know the age difference between groups of time since last menstruation in figure 3. However, authors answered well to questions.